# Variation in phenotypes from a Bmp-Gata3 genetic pathway is modulated by Shh signaling

**Mary E. Swartz** [ID]*, **C. Ben Lovely** [ID]¤, **Johann K. Eberhart** [ID]

Department of Molecular Biosciences, University of Texas at Austin, Austin, Texas, United States of America

¤ Current Address: Department of Biochemistry and Molecular Genetics, University of Louisville, Louisville, Kentucky, United States of America
* swartz@austin.utexas.edu

**Data Availability Statement:** All relevant data are within the manuscript and its Supporting information files.

**Funding:** This work was supported by (NIDCR https://www.nidcr.nih.gov) RO1 DE020884 and

## Abstract

We sought to understand how perturbation of signaling pathways and their targets generates variable phenotypes. In humans, *GATA3* associates with highly variable defects, such as HDR syndrome, microsomia and choanal atresia. We previously characterized a zebrafish point mutation in *gata3* with highly variable craniofacial defects to the posterior palate. This variability could be due to residual Gata3 function, however, we observe the same phenotypic variability in *gata3* null mutants. Using *hsp*:*GATA3-GFP* transgenics, we demonstrate that Gata3 function is required between 24 and 30 hpf. At this time maxillary neural crest cells fated to generate the palate express *gata3*. Transplantation experiments show that neural crest cells require Gata3 function for palatal development. Via a candidate approach, we determined if Bmp signaling was upstream of *gata3* and if this pathway explained the mutant's phenotypic variation. Using *BRE*:*d2EGFP* transgenics, we demonstrate that maxillary neural crest cells are Bmp responsive by 24 hpf. We find that *gata3* expression in maxillary neural crest requires Bmp signaling and that blocking Bmp signaling, in *hsp*:*DN-Bmpr1a-GFP* embryos, can phenocopy *gata3* mutants. Palatal defects are rescued in *hsp*:*DN-Bmpr1a-GFP;hsp*:*GATA3-GFP* double transgenic embryos, collectively demonstrating that *gata3* is downstream of Bmp signaling. However, Bmp attenuation does not alter phenotypic variability in *gata3* loss-of-function embryos, implicating a different pathway. Due to phenotypes observed in hypomorphic *shha* mutants, the Sonic Hedgehog (Shh) pathway was a promising candidate for this pathway. Small molecule activators and inhibitors of the Shh pathway lessen and exacerbate, respectively, the phenotypic severity of *gata3* mutants. Importantly, inhibition of Shh can cause *gata3* haploinsufficiency, as observed in humans. We find that *gata3* mutants in a less expressive genetic background have a compensatory upregulation of Shh signaling. These results demonstrate that the level of Shh signaling can modulate the phenotypes observed in *gata3* mutants.

R35 DE029086 to JKE, (NIH/NIAAA https://www.
niaaa.nih.gov) F32AA021320 and K99AA023560 to
CBL The funders had no role in study design, data
collection and analysis, decision to publish, or
preparation of the manuscript.

**Competing interests:** The authors have declared
that no competing interests exist.

## Author summary

Human birth defects vary widely in their presentation. This is true even in cases where the
underlying genetic mutation is the same. In humans, mutation of the gene *GATA3* associates with two highly variable birth defects that can disrupt development of the face, microsomia and Hypoparathyroidism, Deafness and Renal dysplasia (HDR) syndrome. We
used the zebrafish to identify the causes of variation in facial defects associated with *gata3*.
We show that the cells that generate the palate require the function of Gata3 and that the
Bone Morphogenetic Protein (Bmp) pathway is necessary for the expression of *gata3* by
these cells. While Gata3 functions downstream of Bmp, we find no evidence that alteration of the Bmp pathway causes the variability in skeletal defects in *gata3* mutants.
Instead, we identify a separate signaling pathway, the Sonic Hedgehog (Shh), pathway that
is responsible for the variability in *gata3* mutant defects. In a genetic background that promotes mild *gata3* mutant phenotypes, Shh signaling is elevated relative to mutants in a
genetic background sensitized for severe defects. Reduction or elevation of Shh signaling
in these two mutants, exacerbates and lessens the phenotypic severity, respectively. Thus,
our finding provides important insight into how interactions between signaling pathways
cause variation in human birth defects.

## Introduction

Congenital birth defects are a leading cause of infant mortality worldwide and the leading
cause of mortality in many industrial nations according to the World Health Organization.
The causes of most birth defects are thought to be complex and include genetic and environmental risk factors. Furthermore, the precise phenotypes observed within a specific birth
defect can be highly variable and this variability is also thought to arise from genetic and environmental modifiers. Craniofacial defects are among the most common birth defects and offer
an excellent model of variability. For instance, orofacial clefts affect 1 in 700 live births and
appear to be caused by an interplay of genetic and environmental factors [1].

The high rate and variable nature of craniofacial defects such as orofacial clefts are largely
because proper palatogenesis requires the precise coordination of many events that are subject to genetic and/or environmental perturbations. Cranial neural crest cells (CNCC) that
generate the palatal skeleton are generated in the dorsal neural tube from which they must
migrate into the periphery to differentiate. Palatal precursors occupy the maxillary region of
the first pharyngeal arch and the frontonasal prominence in human, mouse and zebrafish
[1–4]. The zebrafish palate (also referred to as the anterior neurocranium) is comprised of
an anterior, midline, ethmoid plate and the posterior bilateral trabeculae. Fate mapping
shows that the medial ethmoid palate is formed from frontonasal CNCC and the remaining
palate forms from maxillary CNCC [2,3]. The trabeculae fuse to the posterior neurocranium
which is primarily composed of mesodermally-derived cells [5]. While the evolutionary
homologies remain unclear, a growing body of evidence demonstrates that the gene function required for craniofacial development, including palatogenesis, in mammals is conserved in zebrafish [6–9]. Yet we still have an incomplete knowledge of the genes involved
in craniofacial development and a poor understanding of how they interact to generate
variability.

Defining the causes of phenotypic variation is important for our understanding of development, disease and evolution. However, there are a limited number of studies defining the
cause of phenotypic variation. Such variation can conceptually be caused by three general

mechanisms: 1) genetic background, 2) gene-environment interactions and 3) stochastic developmental events [10]. Our understanding of all of these mechanisms is limited. Recent work is beginning to shed light on the mechanisms of gene-environment interactions and how such interactions can synergistically effect phenotypes [11,12]. Similarly, mutant analyses in mouse and zebrafish have pointed to the importance of genetic background, with phenotypes differing depending upon the strain carrying the mutation [13–16]. Recent studies have demonstrated that selective breeding for heritable variation in phenotypic penetrance of *mef2ca* mutants results in altered methylation in a transposon at the *mef2ca* promoter and a compensatory downregulation of the opposing Notch pathway [17,18], providing some insights into these mechanisms. However, much remains to be understood regarding the nature of these genetic background effects.

The zinc finger transcription factor, GATA3, associates with craniofacial syndromes. Haploinsufficiency of *GATA3* causes Hypoparathyroidism, Deafness and Renal dysplasia (HDR) syndrome [19]. HDR is an extremely variable birth defect, even among individuals sharing the same mutation within a family [20]. Palatal defects and choanal atresia (defects of the nasal bones) are craniofacial defects that can co-occur with the HDR triad [21,22]. Furthermore, *GATA3* associates with craniofacial microsomia [23–25], another highly variable disease. Humans with microsomia can have unilateral shortening and clefts of the palate as well as defects to other craniofacial bones, ears and cranial ganglia [24,26]. While HDR is a relatively rare disease microsomia is very common affecting 1 in 5600 conceptuses. Collectively, these findings in human patients suggest that phenotypes associated with loss of GATA3 function are inherently variable.

Our understanding of the roles of GATA3 in craniofacial development is limited due to the early lethality of mouse *Gata3* mutants caused by parathyroid defects [27,28]. *Gata3* mutant mice pharmacologically rescued display severe craniofacial defects and neural crest patterning defects [28,29], consistent with a critical role in craniofacial development. Work in the mouse mandible has demonstrated that Smad1/5 binds regions adjacent to *Gata3*, suggesting that it is a target of Bmp signaling [30] furthermore Gata3 is a BMP target in limb mesenchyme [31]. However, the precise roles of *Gata3*, its regulation during palate development and the modulation of resulting phenotypes remain unknown.

We previously demonstrated that phenotypes in zebrafish *gata3* mutants were highly variable, similar to human disorders associated with *GATA3*, and that this variability associated with genetic background [13]. Here, we determine the role of *gata3* in development of the zebrafish palate and characterize the signaling pathways that regulate the variability in *gata3* mutant phenotypes. We show that neural crest cells require the function of Gata3 shortly after their migration into the pharyngeal arches. Bmp signaling is necessary for the expression of *gata3* in the maxillary neural crest and loss of Bmp signaling recapitulates the craniofacial defects in *gata3* mutants. Transgenic overexpression of *GATA3* restores facial development in Bmp deficient zebrafish, demonstrating that Gata3 functions downstream of Bmp. We demonstrate that the variability in *gata3* mutant phenotypes is due to the actions of a second pathway, Shh. Elevating and attenuating Shh signaling ameliorates and exacerbates the phenotypes of *gata3* mutants, respectively. Importantly, in a sensitized *gata3* mutant genetic background, reduction of Shh signaling is sufficient to cause *gata3* haploinsufficiency, similar to the human condition. Our results demonstrate that the coordination of two pathways, a Bmp-Gata3 pathway and Shh, regulate trabeculae phenotypes. These findings provide important insights into the causes of variability in craniofacial disease phenotypes.

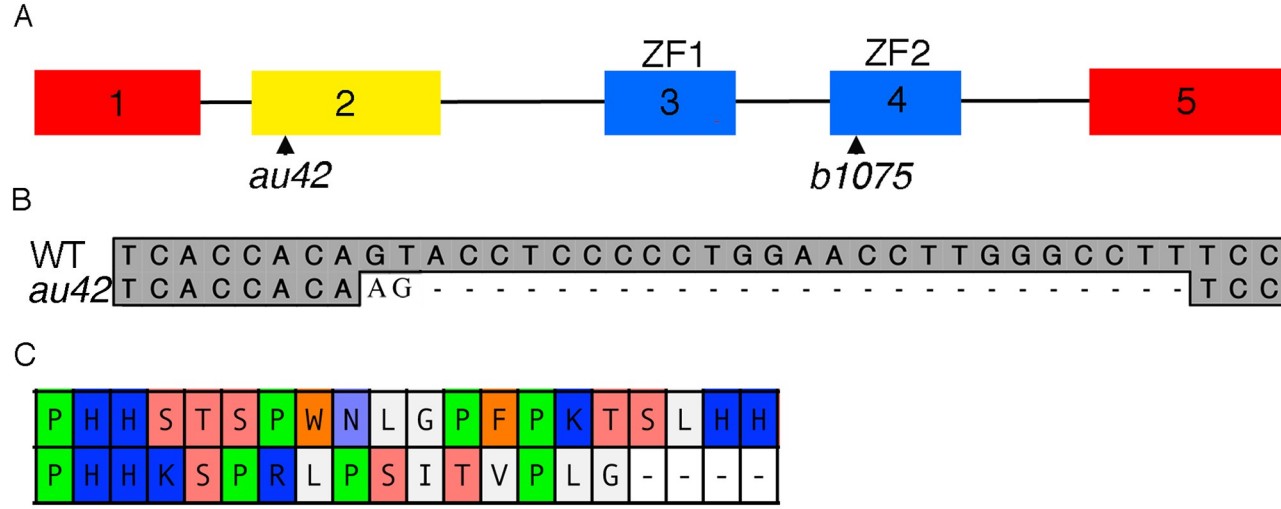

**Fig 1. A null allele of *gata3* generated via CRISPR-Cas9.** (A) As previously reported *gata3*[b1075] mutation is located in exon 4. The CRISPR *gata3* mutant allele *au42* disrupts the second exon. (B) The *gata3 au42* allele is a two base pair insertion/25 base pair deletion. (C) Predicted wild type and *gata3*[au42] mutant protein sequences indicates a frame shift followed by four consecutive stops in translation.

## Results

### Loss of Gata3 results in highly unstable palatal phenotypes

Previously, we reported a missense mutant allele of *gata3*, *b1075*, [13] (Fig 1A) that displayed highly variable palatal phenotypes. We demonstrated that selective breeding across genetic backgrounds, could be used to select for separate populations that consistently produce phenotypes at each end of this spectrum. These phenotypes ranged from a severe truncation of the trabeculae, resulting in the failure of trabeculae to fuse to the mesoderm-derived posterior neurocranium (which we refer to as a cleft for simplicity) in one population, to mild cell rearrangements within the trabeculae, in a second population [13]. The *gata3*[b1075] allele disrupts a cysteine that coordinates the zinc ion in the DNA-binding zinc finger of Gata3. The nature of this mutation mirrored a human *GATA3* mutation demonstrated to lack DNA binding capability in cell culture [32]. However, Gata transcription factors have been shown to effect gene expression without DNA binding, via protein-protein interactions [33]. Thus, it remained plausible that the variability in the zebrafish mutant was due to residual Gata3 function within an *in vivo* context.

We generated a new allele of *gata3* using CRISPR/Cas-9. We targeted the second exon of *gata3* to delete both zinc finger domains and generated the allele, *au42*, that has a two base-pair insertion followed by a 25 base pair deletion (Fig 1A and 1B). By sequencing mutant mRNA, we confirmed the indel and identified four in frame stop codons following the indel (Fig 1C). The predicted truncated protein lacks both zinc finger domains, which are essential for function. Thus, we conclude that *au42* is a null allele of *gata3*.

Interestingly, *gata3*[au42] mutants exhibited the full spectrum of phenotypes observed in *gata3*[b1075] mutants (Fig 2; [13]. In/del mutations, such as *au42*, can be associated with nonsense-mediated decay, which can result in genetic compensation [34]. However, the *b1075* allele is a mis-sense mutation, unlikely to undergo nonsense mediated decay. Thus, the similarity in phenotypes suggests a mechanism other than genetic compensation following nonsense mediated decay for the phenotypic variability.

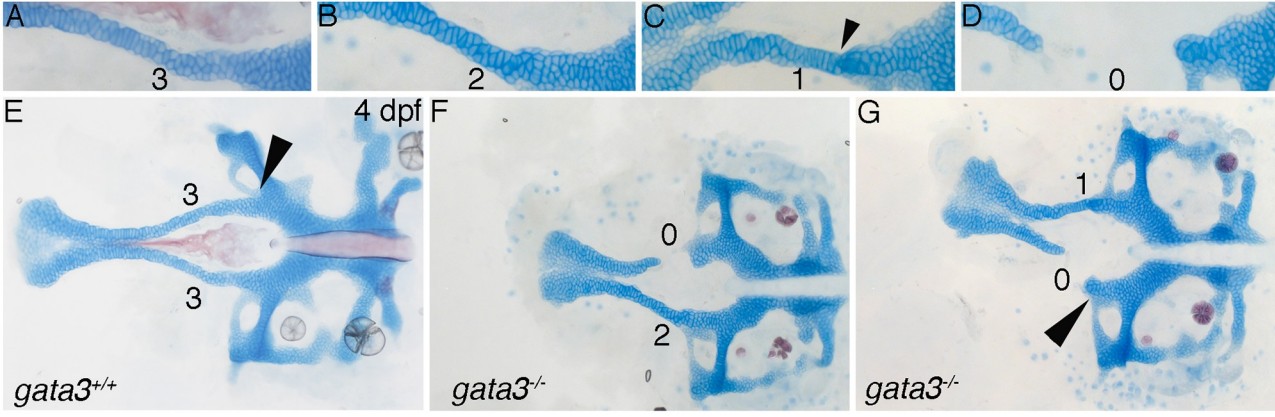

**Fig 2. Quantitative scoring of *gata3* mutant phenotypes.** (A-G) Flat mounts of 4 dpf neurocrania, anterior to the left showing magnified images of trabeculae. Phenotypes were scored on individual sides of the neurocranium according to the level of disruption to the trabeculae. (A) Wild type phenotype (score = 3) with trabeculae fused and in one plane with the posterior neurocranium. The cells within the trabeculae are arranged into a column resembling a stack of coins. (B) Disruption to the columnar arrangement of trabeculae cells (score = 2) with proper fusion to the posterior neurocranium with all cells in one plane. (C) Trabeculae cells not stacked properly and fused inappropriately, below the plane of the posterior neurocranium, see arrowhead (score = 1). (D) Loss of trabeculae cells (score = 0) and therefore no fusion to the posterior neurocranium. (E-G) 4 dpf flat mounted neurocrania, anterior to the left showing examples of trabeculae scoring. (E) Wild type zebrafish with arrowhead indicating proper lateral commissure fusion. (F-G) *gata3* mutant embryos, arrowhead in G indicating improper lateral commissure fusion. In subsequent figures the numbering above the trabeculae reflect trabeculae scoring.

As with *gata3^{b1075}* mutants, *gata3^{au42}* mutant craniofacial phenotypes appear restricted to the trabeculae of the zebrafish palate and the lateral commissure. In zebrafish with the least severe phenotype the trabeculae fuses appropriately to the posterior neurocranium but is shortened and the cells within the trabeculae are inappropriately stacked but are in the same plane as and fused to the posterior neurocranium (Fig 2B and 2F; for our quantification below this is scored as a 2). We observed an intermediate phenotype in which the trabeculae are intact but have inappropriately stacked cells that fuse incorrectly to the ventral side of the posterior neurocranium so that the trabeculae and the posterior neurocranium are not in the same plane. (Fig 2C, arrowhead, and 2G; scored as a 1). The most severe phenotype is the cleft of the trabeculae in which the trabeculae are shortened and do not fuse to the posterior neurocranium (Fig 2D, 2F and 2G; scored as a 0). Despite the variability in the trabeculae phenotypes, which we focus on here, we note that mutants are 100% penetrant for a defect in which the lateral commissure fuses inappropriately to a more anterior region of the neurocranium. Compare arrowhead in Fig 2E indicating the wild type fusion of lateral commissure cell to arrowhead in Fig 2G indicating improper fusion.

We further characterized the nature of the phenotypic variability by comparing phenotypes within larvae. We compared the score from the left and right trabeculae of 39 neurocrania (Table 1). Neurocrania with concordant scores (green boxes) are slightly less numerous (n = 18) than those neurocrania that have discordant scores (blue and yellow boxes, n = 21). Neurocrania scores differing by more than one (yellow boxes) are uncommon. This asymmetry fluctuates in its sidedness. However, the left side tends to have a less severe phenotype.

## Neural crest cells require the function of Gata3

To begin to understand how Gata3 functions during palatal development we assayed its spatiotemporal expression during craniofacial development. Riboprobes against *gata3* label the head by 22 hpf (Fig 3A). However, at this time the expression appears to be in neural progenitors (black arrow), the ear (e) and what is likely to be endoderm (asterisk). Neural crest expression

**Table 1. Contralateral score.**

| | | Right side | | | |
|---|---|---|---|---|---|
| | | 0 | 1 | 2 | 3 |
| **Left side** | 0 | 2 | 4 | 0 | 0 |
| | 1 | 7 | 12 | 3 | 0 |
| | 2 | 1 | 6 | 4 | 0 |
| | 3 | 0 | 0 | 0 | 0 |

of *gata3* begins by 24 hpf and continues until 48 hpf (Fig 3B–3D). At 24 hpf, neural crest cells just ventral to the eye begin expressing *gata3* transcripts and this staining becomes prominent by 26 hpf (Fig 3B and 3C, white arrowheads). This expression domain matches the region fate mapped to become the anterior neurocranium, particularly the trabeculae, via single cell fate mapping at 24 hpf [2,3] and kaede photoconversion at 36 hpf [6]. By 48 hpf, when the progenitors of the trabeculae have extended into their rod-shaped structure [2,6], the expression of *gata3* becomes restricted away from these precursors (Fig 3D). To verify that maxillary neural crest cells express *gata3* we performed fluorescent in situ hybridization on 36 hpf embryos using probes to *gata3* and *pdgfra*, as a marker of neural crest cells. The overlap of *gata3* and *pdgfra* confirms that *gata3* is expressed in maxillary crest cells. Thus, the expression pattern of *gata3* in the early maxillary domain suggests a role in the cell behaviors that mediate extension of the zebrafish palatal skeleton.

Given the dynamic nature of the expression of *gata3*, we sought to determine when palatal development requires Gata3 function. We generated a *hsp*:GATA3-EGFP heat shock transgenic line to temporally regulate the expression of *GATA3* (Fig 4). Three hours following heat shock, the GATA3-EGFP fusion protein clearly localized to cell nuclei (Fig 4A), consistent with it being a functional fusion protein. To determine when palate development requires GATA3 function, we determined when transgenic expression of *GATA3-EGFP* could rescue the phenotype of *gata3*[au42] mutants.

Embryos were heat shocked at times points ranging from 18 hpf, as a control for a time point prior to neural crest expression of *gata3*, to 48 hpf and then grown up to 4 dpf. All embryos were stained and trabeculae on each side of the palate were scored according to the system outlined in Fig 2. We found that ectopic expression of *GATA3* had no apparent effect on the phenotypes of embryos wild type for *gata3* (Fig 4B) and (S1 Fig). Heat shock prior to 24 hpf did not improve the development of the trabeculae (Fig 4D). The trabeculae phenotype was most significantly rescued by heat shock at 24 hpf (Fig 4C and 4D), although there was some rescue as late as 30 hpf. See Table 2 for a complete list of statistics. Thus, Gata3 appears to function over a small window of time when maxillary neural crest cells are condensing in the pharyngeal arches to promote trabeculae development.

Maxillary neural crest cells express *gata3* during the period when it is required for trabeculae development. To determine if cranial neural crest cells require *gata3* autonomously, we created genetic chimeras. We transplanted membrane labeled *sox10:mCherry* CNCC into *fli1*:*EGFP;gata3* mutant hosts unilaterally (Fig 5). This allowed us to compare the phenotype on the control side, comprised of all mutant cells, to the transplanted side, chimeric for mutant and wild type cells. Embryos were imaged at 30 hpf (Fig 5A) to determine the contribution of the transplanted cells. Those embryos with sizeable contribution to maxillary neural crest cells were then grown to 4 dpf and stained for cartilage and bone (Fig 5B, n = 10). The neurocrania were flat mounted and the trabeculae were scored as above. A Mann-Whitney test demonstrated that the presence of wild type neural crest cells significantly restored the trabeculae (Fig 5C, p<0.05). Of the 10 transplanted embryos six were chimeric on the left and four were

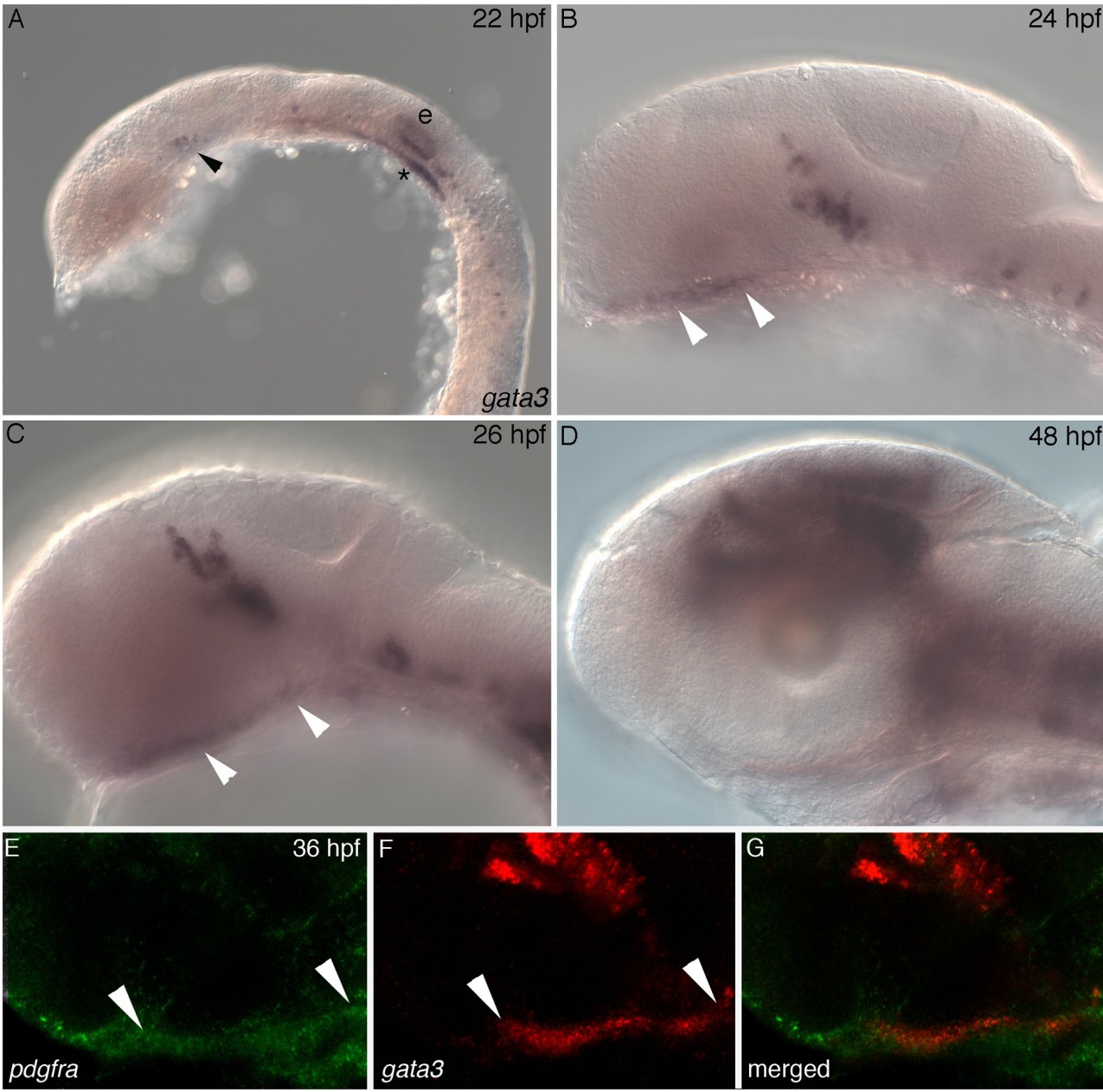

**Fig 3. Dynamic expression of *gata3* in maxillary neural crest cells.** (A-D) Lateral views, anterior to the left, of *gata3 in situ* hybridization. (A) At 22 hpf, *gata3* is expressed in the head in brain (black arrow), ear (e) and what is likely to be endoderm (*). (B) By 24 hpf, *gata3* expression domains include cranial neural crest cells, which contain the trabeculae precursors (white arrowheads), the brain and neurons. (C) At 26 hpf expression continues in all previously expressing tissues. The white arrowheads indicate the cranial neural crest cells that give rise to trabeculae. (D) At 48 hpf expression begins to be down regulated in the developing palate. (E-G) RNA Scope V2 whole mount *in situ* hybridization at 36 hpf; white arrowheads bracket the trabeculae precursors. (E) Expression of *pdgfra* in cranial neural crest cells. (F) Expression of *gata3* in maxillary cranial neural crest cells that will give rise to trabeculae. (G) The overlap of of *pdgfra* and *gata3* in trabeculae precursors.

chimeric on the right. While there is often phenotypic variability within an individual *gata3* mutant, we find that the phenotypic score on the transplanted side was always improved relative to the non-transplanted side. To determine if donor neural crest cells contribute to the rescued trabeculae, we transplanted cells from *ubi*:*Switch* (ubiquitously labeling cells green in the

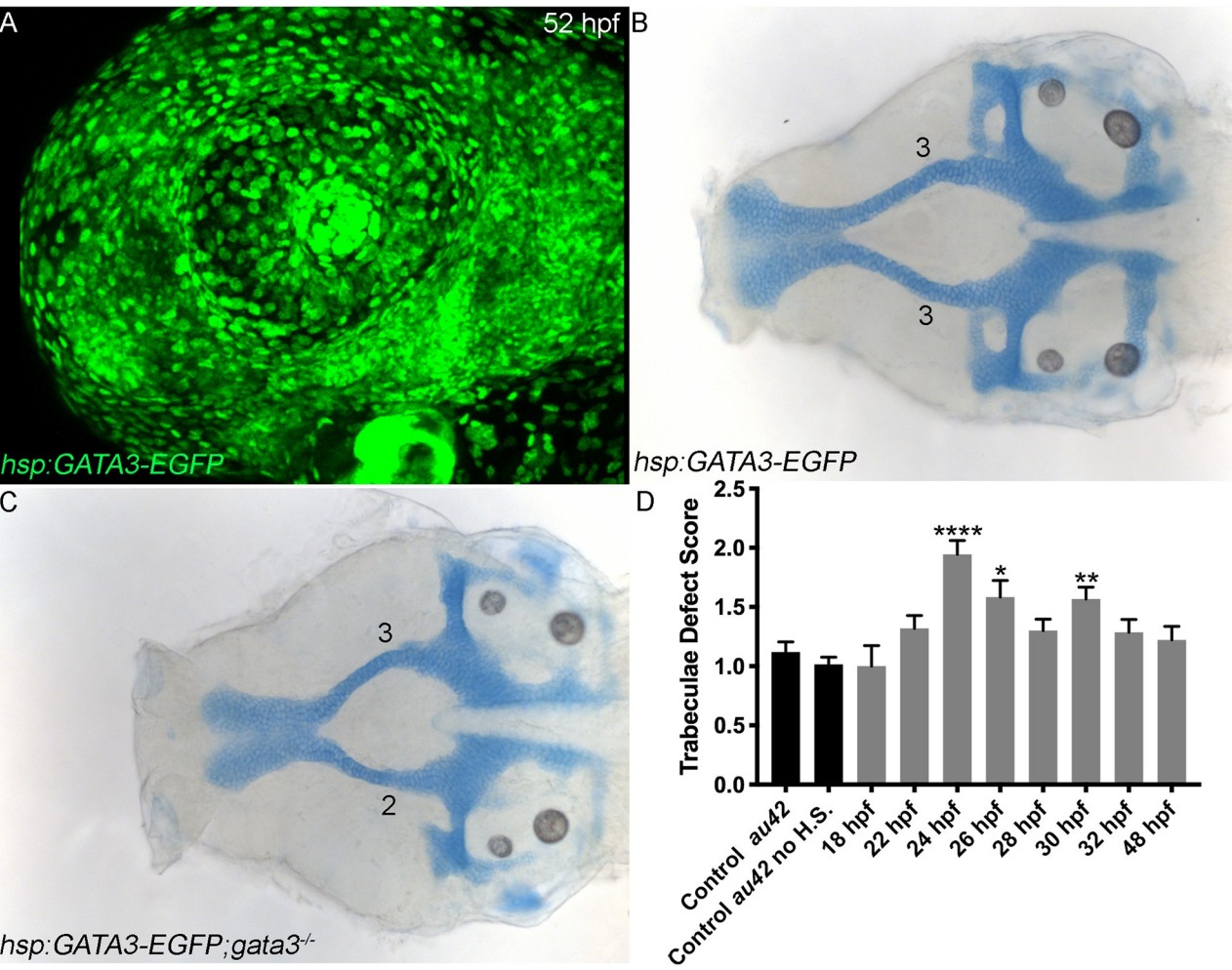

**Fig 4. Gata3 function is required during early palatogenesis.** (A) Embryo carrying the *hsp*:*GATA3-EGFP* transgene heat shocked at 48–49 hpf and imaged at 52 hpf expressing human *GATA3* in every cell. (B) Flat mount of a cartilage stained 4 dpf *hsp*:*GATA3-EGFP*;*gata3$^{+/+}$* embryo that was heat shocked at 24 hpf with no resulting visible malformations. (C) Flat mount of a cartilage stained 4 dpf neurocranium carrying *hsp*:*GATA3-EGFP*; *gata3$^{au42/au42}$*. The mutant embryo was heat shocked at 24 hpf and resulted in a substantial rescue of the trabeculae phenotype. (D) Graph depicting quantification of mutant *gata3$^{au42}$* phenotype scores at different heat shock time points. Trabeculae phenotype is rescued significantly by *hsp*: *GATA3-EGFP* at 24–26 hpf and also at 30 hpf. A one-way ANOVA was performed followed by Dunnett's multiple comparisons test, *gata3$^{au42/au42}$* control was compared to all other conditions.

absence of Cre) into *sox10:mCherry* hosts injected with *gata3* morpholino. We find that these transplanted cells are capable of contributing to the trabeculae and that the trabeculae is rescued on the transplanted side (Fig 5F–5H). These results support a model in which neural crest cells require the function of *gata3* for proper trabeculae morphogenesis. However, it remains possible that *gata3* is not specifically required within the progenitors of the trabeculae, a genetic fate map of the descendants from thes *gata3*-positive cells will aid in this understanding.

## Gata3 functions downstream of Bmp

We next asked how Gata3 may function in the development of such a specific part of the skull and why the phenotype can be so highly variable. While blocking Bmp signaling disrupts multiple aspects of craniofacial development, similar trabeculae defects can be generated via a

**Table 2. *gata3* rescue statistics.**

| Controls | x̄ | n | S.E.M. | P-value |
|---|---|---|---|---|
| *gata3*⁻ᐟ⁻ | 1.1 | 84 | | |
| *hsp*:*GATA3-EGFP*;*gata3*⁻ᐟ⁻ no heat shock | 1.0 | 124 | 0.1118 | |
| Hpf of heat shock *hsp*:*GATA3-EGFP*;*gata3*⁻ᐟ⁻ | | | | |
| 18 | 1.0 | 12 | 0.2441 | n.s. |
| 22 | 1.3 | 53 | 0.1388 | n.s. |
| 24 | 1.9 | 73 | 0.1266 | <0.0001 |
| 26 | 1.6 | 36 | 0.1576 | 0.0245 |
| 28 | 1.3 | 80 | 0.1236 | n.s. |
| 30 | 1.6 | 60 | 0.1337 | 0.0068 |
| 32 | 1.3 | 42 | 0.1495 | n.s. |
| 48 | 1.2 | 36 | 0.1576 | n.s. |

dominant negative form of *bmpr1a* [35]. Furthermore, Smad1/5 binds upstream of *Gata3* in mouse neural crest cells [30]. These results suggest that *gata3* may be a Bmp target in the maxillary neural crest. We performed *in situ* hybridization with *gata3* riboprobe in 36 hpf *smad5*⁻ᐟ⁻ and wild type embryos (Fig 6). We find a striking reduction in the maxillary expression of *gata3* when compared to the wild type sibling (Fig 6, arrowheads). Interestingly, brain and ear expression are mostly unaffected (Fig 6). These data demonstrate that the neural crest expression of *gata3* requires Bmp signaling.

Collectively, these findings suggest an epistatic relationship in which Bmp signaling to the neural crest is upstream of *gata3*. This predicts that maxillary neural crest will be Bmp responsive. We detected Bmp responsive cells using *BRE*:*d2EGFP* transgenics, in which a destabilized form of GFP is expressed from a Bmp response element. We crossed this line to the *sox10*: *mRFP* line, allowing us to visualize the overlap of Bmp signaling within the neural crest. Double-labeled embryos were imaged at 24 hpf, the time when *gata3* function is required (Fig 6E–6G). At this time, maxillary neural crest cells display a robust bmp response (Fig 6E and 6G, arrows). Furthermore, using *hsp*:*DN-BmpR1a* transgenics, we demonstrate that the maxillary expression of *gata3* requires Bmp signaling at this time (Fig 6C and 6D). These data support a model in which Bmp signaling is upstream of *gata3*, driving palate development.

This epistatic relationship predicts that forced *gata3* expression should partially compensate for the loss of Bmp signaling. We generated *hsp*:*GATA3-EGFP*;*hsp*:*DN-Bmpr1a* double transgenics to simultaneously express a dominant negative Bmp receptor and GATA3. We heat shocked embryos between 24 to 28 hpf, time points important for *gata3* function, and grew all fish to 4 dpf when they were stained for cartilage and bone then scored for phenotypes (Fig 7). We find that at each time point, co-expression of GATA3 and the DN-Bmp receptor (Fig 7C and 7D) improved the trabeculae phenotype relative to embryos only expressing DN-Bmp receptor (Fig 7B and 7D). The complete list of statistics is in Table 3. As controls for potential non-specific interactions based on the use of two *hsp* promoters, we examined the phenotypes of *hsp*:*DNBmpR1a*;*hsp*:*Gal4* double transgenics. Phenotypes in these embryos mirror those found in *hsp*:*DNBmpR1a* single transgenics (S1 Fig). Whole larvae images of 5 dpf heat shocked, control embryos and *gata3* mutant stained for cartilage and bone are also in S1 Fig. Additionally, we characterized fluorescent intensity at the cell membrane in *hsp*:*DNBmpR1a* and *hsp*:*DNBmpR1a*;*hsp*:*GATA3-EGFP*. We imaged periderm cells due to their large, flat morphology making it easy to distinguish cell membrane (DNBmpR1a) and nuclear (GATA3) labeling. We find no difference in fluorescent intensity between the single and double

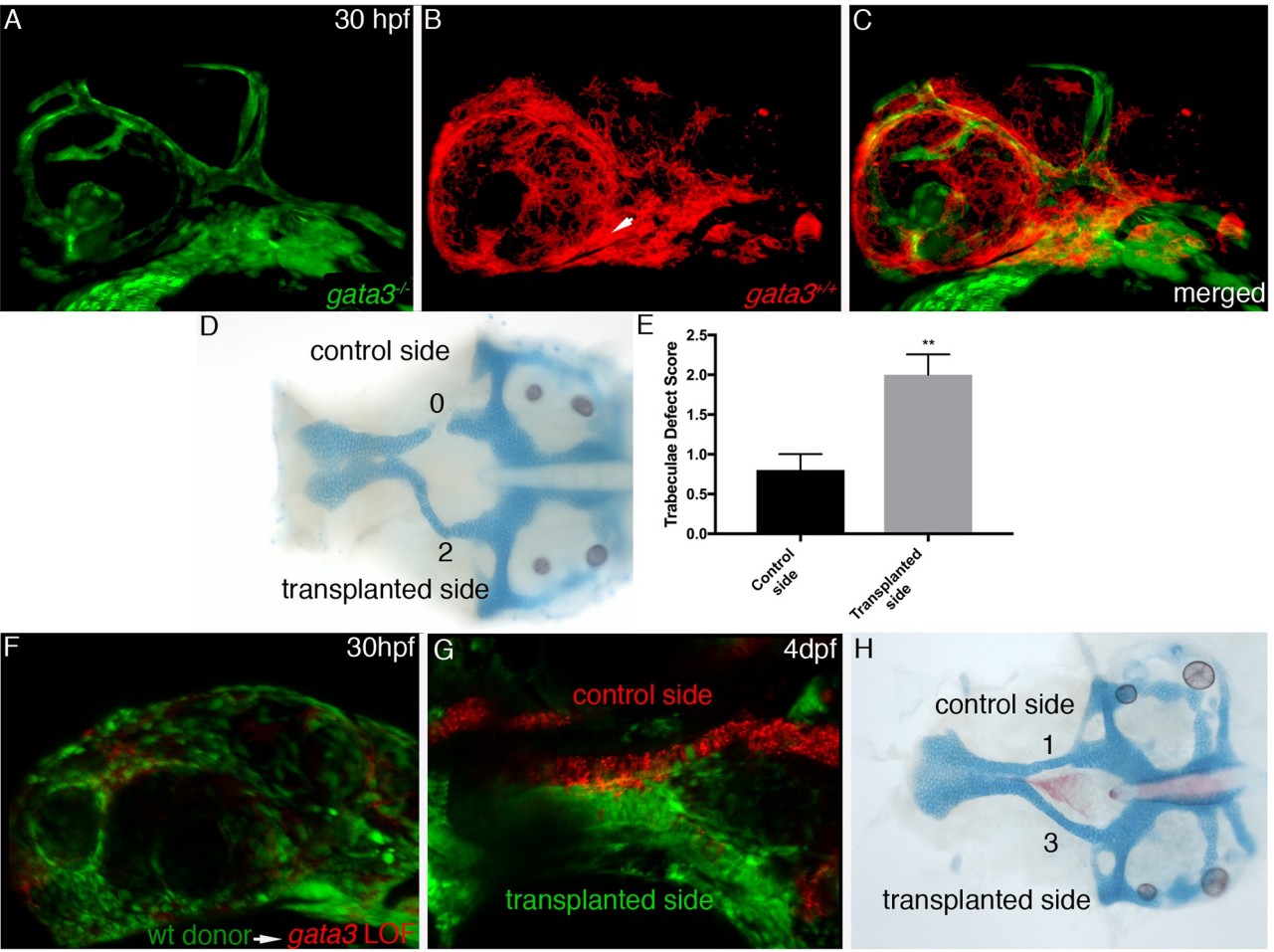

**Fig 5. Chimeric analyses demonstrate that cranial neural crest require *gata3*.** (A-C) Wild type *sox10:mCherry* CNCC (in red) were transplanted unilaterally into *gata3;fli1:EGFP* mutant hosts (in green). The resulting embryo was imaged at 30 hpf showing the major contribution to the maxillary is wild type (arrow). (D) Flat mount of 4dpf neurocrania of the same fish showing the rescue of the trabeculae (arrow) compared to the loss of trabeculae on the non-transplanted side of the embryo. (E) Graph depicting the overall trabeculae score of the 10 chimeric embryos with the transplanted sides phenotype significantly improved relative to the non-transplanted sides (Mann-Whitney test P = 0.0052). (F) Wild type *ubi:Switch* (in green) were transplanted unilaterally into *gata3* LOF;*sox10:mCherry*. Chimeric embryos (n = 4) were imaged at 30 hpf to demonstrate donor cells populating the majority of the maxillary domain. (G) Flat mount of same fish at 4 dpf showing that the green donor cells populate the trabeculae on the transplanted side. (H) Stained flat mount of same embryo at 4 dpf showing complete rescue of the trabeculae phenotype. Compare the transplanted trabeculae to the disorganized and improperly fused trabeculae on the control side.

transgenics (S2 Fig). Taken together our results place *gata3* downstream of Bmp signaling with both being required for proper trabeculae morphogenesis.

The Bmp pathway has many targets in the neural crest. One possible explanation for the phenotypic plasticity in *gata3* mutants is variable compensation by other Bmp targets. This model predicts that attenuation of Bmp signaling in a *gata3* loss-of-function embryo will cause more severe phenotypes. To test this, we injected a *gata3* morpholino that faithfully recapitulated the phenotypes observed in our *gata3^{au42}* null mutant into embryos heterozygous for *smad5*. However, the phenotypes in *gata3* morpholino-injected wild type and *smad5* heterozygous embryos were identical (p = 0.152, wild type mean = 0.87, S.E.M. = 0.1 n = 52; heterozygotes mean = 0.67, S.E.M. = 0.09, n = 72). While these findings do not rule out the possibility that other Bmp targets modulate the phenotype of *gata3* mutants, they suggest that *gata3* is a critical target.

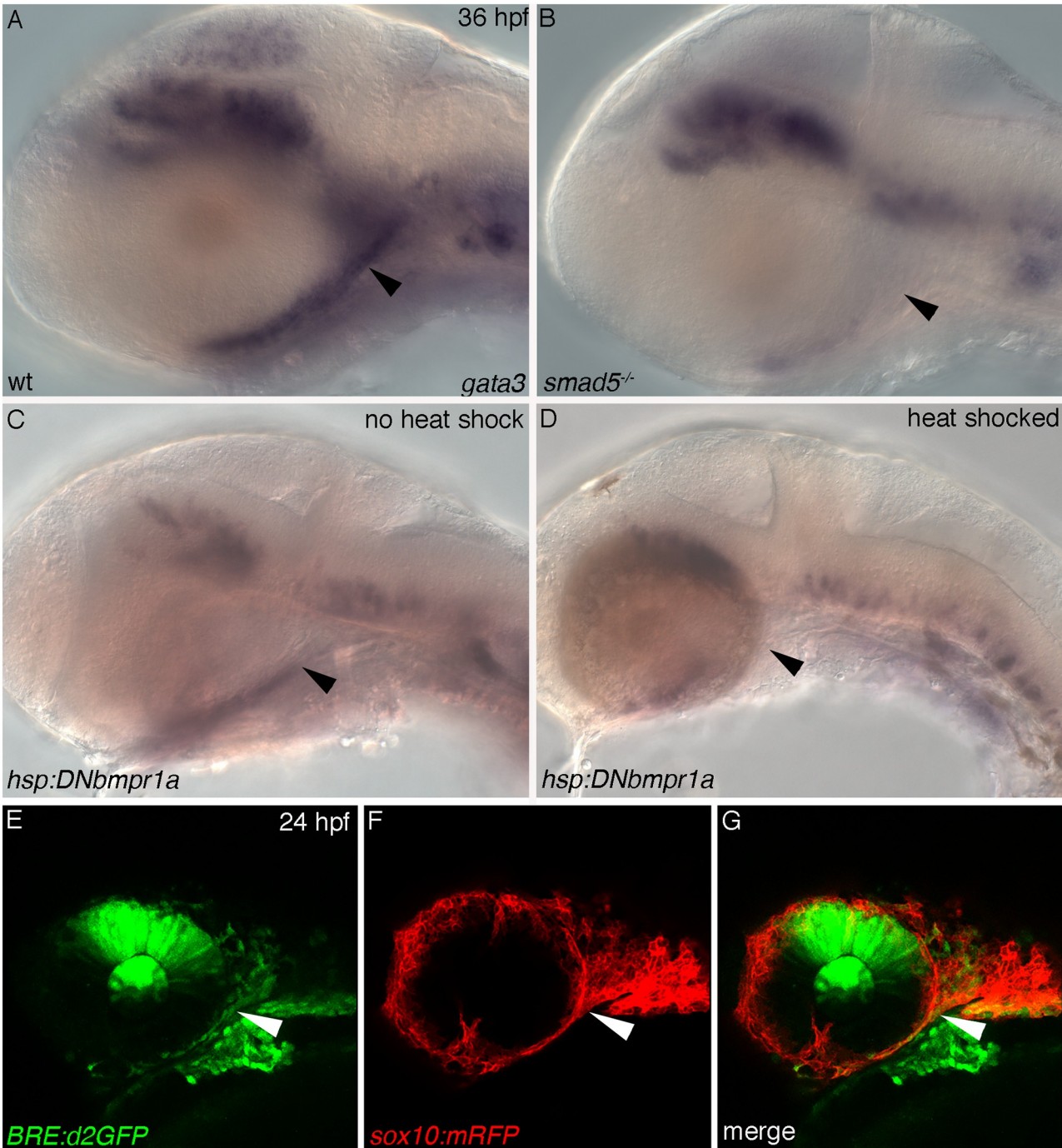

**Fig 6. Bmp signaling to maxillary neural crest regulates the expression of *gata3*.** (A-D) Lateral views with anterior to the left of 36 hpf *gata3* in situ hybridization black arrows point to maxillary crest that will give rise to trabeculae; white arrows mark the ear. (A-B) Embryos from a *smad5* clutch. (A) Wild type embryo showing strong expression of *gata3* in maxillary trabeculae precursors. (B) Mutant embryo showing loss of *gata3* expression in trabeculae precursors. (C-D) Embryos from a *hsp:DN-Bmpr1a* clutch. (C) No heat shock embryo arrowhead indicates *gata3* expression in maxillary precursors. (D) Embryo heat shocked at 24 hpf arrowhead indicates loss of *gata3* expression in maxillary CNCC. (E-G) Confocal images of Bmp responsive cells (green, cytoplasmic) and CNCC (red, membrane tagged), lateral views with anterior to the left at 24 hpf. Arrow points to Bmp responsive maxillary neural crest cells.

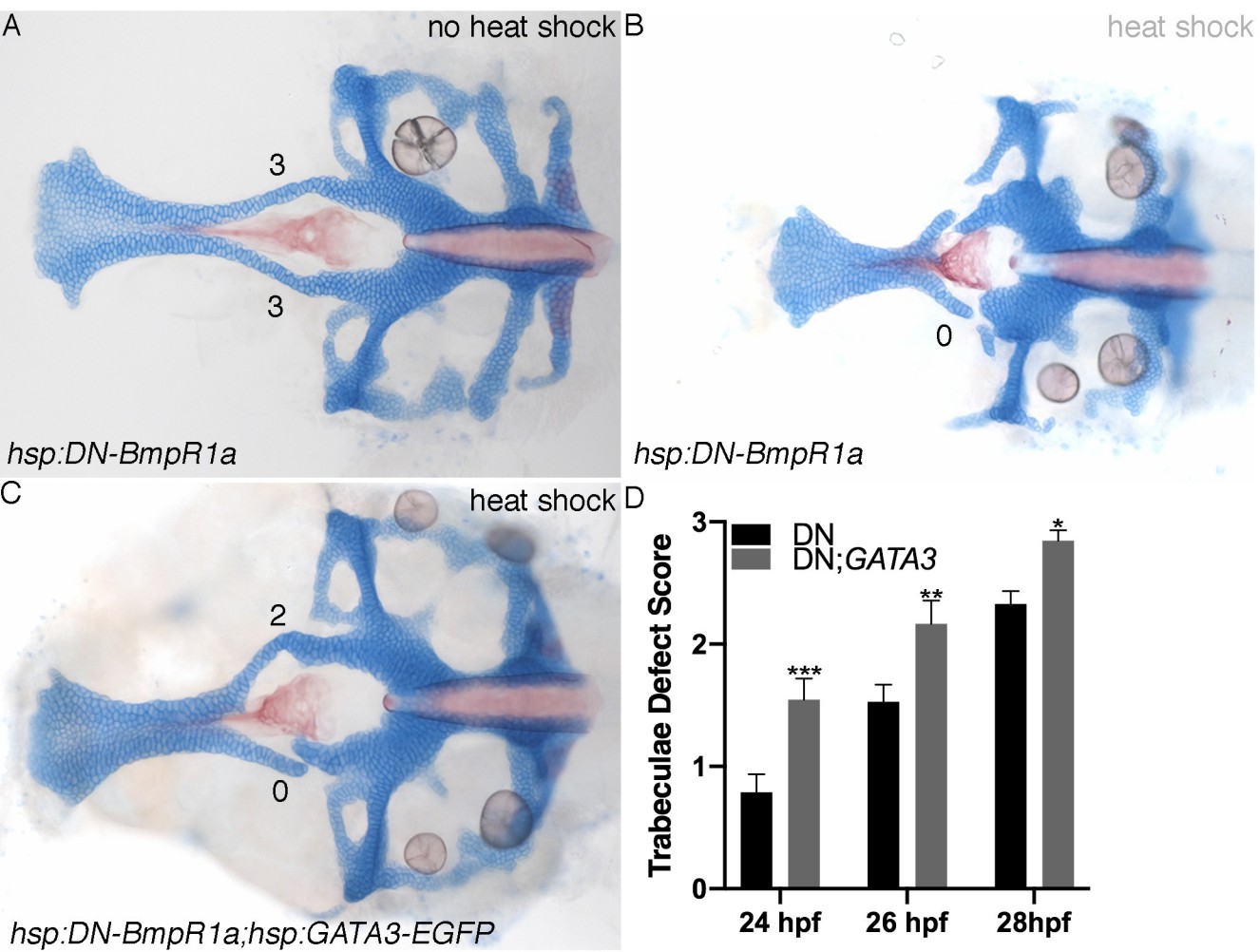

**Fig 7. Gata3 functions downstream of Bmp in palate development.** (A-C) Flat mounts of 4 dpf neurocrania anterior to the left. (A) Wild type phenotype of the *hsp:DN-BmpR1a* with no heat shock. (B) Severely affected trabeculae typical of the *hsp:DN-BmpR1a* when heat shocked at 24 hpf. (C) A typical rescue of the *hsp:DN-BmpR1a;hsp:GATA3-EGFP* heat shock phenotype at 24 hpf. (D) Graph depicting quantification of heat shock phenotype scores at three different heat shock time points. Trabeculae phenotype is rescued significantly by *hsp:GATA3-EGFP* at 24 and 26 hpf. Phenotypes are also rescued at 28 hpf, but to a lesser degree as the overall phenotype is not as severe. A two way ANOVA was performed with Holm-Sidak's multiple comparisons test. At all heat shock timepoints the *hsp:DN-BmpR1a* controls had significantly worse phenotype when compared to the *hsp:DN-BmpR1a;hsp:GATA3-EGFP* double heat shock.

## Shh signaling modulates the phenotypes in *gata3* loss-of-function embryos

In our previous analyses, we found that Shh signaling was critical for development of the zebrafish palate [2]. In those hypomorphic *shha* mutants that retained the palate we often found

**Table 3. Statistics for rescue of *bmp*.**

| Hpf of heat shock | | x̄ | n | S.E.M. | P-value |
|---|---|---|---|---|---|
| 24 | *hsp:DN-Bmpr1a* | 0.79 | 38 | 0.1468 | 0.0007 |
| | *hsp:DN-Bmpr1a;hsp:GATA3-EGFP* | 1.55 | 44 | 0.1734 | |
| 26 | *hsp:DN-Bmpr1a* | 1.53 | 68 | 0.1397 | 0.0082 |
| | *hsp:DN-Bmpr1a;hsp:GATA3-EGFP* | 2.17 | 42 | 0.1895 | |
| 28 | *hsp:DN-Bmpr1a* | 2.32 | 64 | 0.1406 | 0.0131 |
| | *hsp:DN-Bmpr1a;hsp:GATA3-EGFP* | 2.84 | 52 | 0.0841 | |

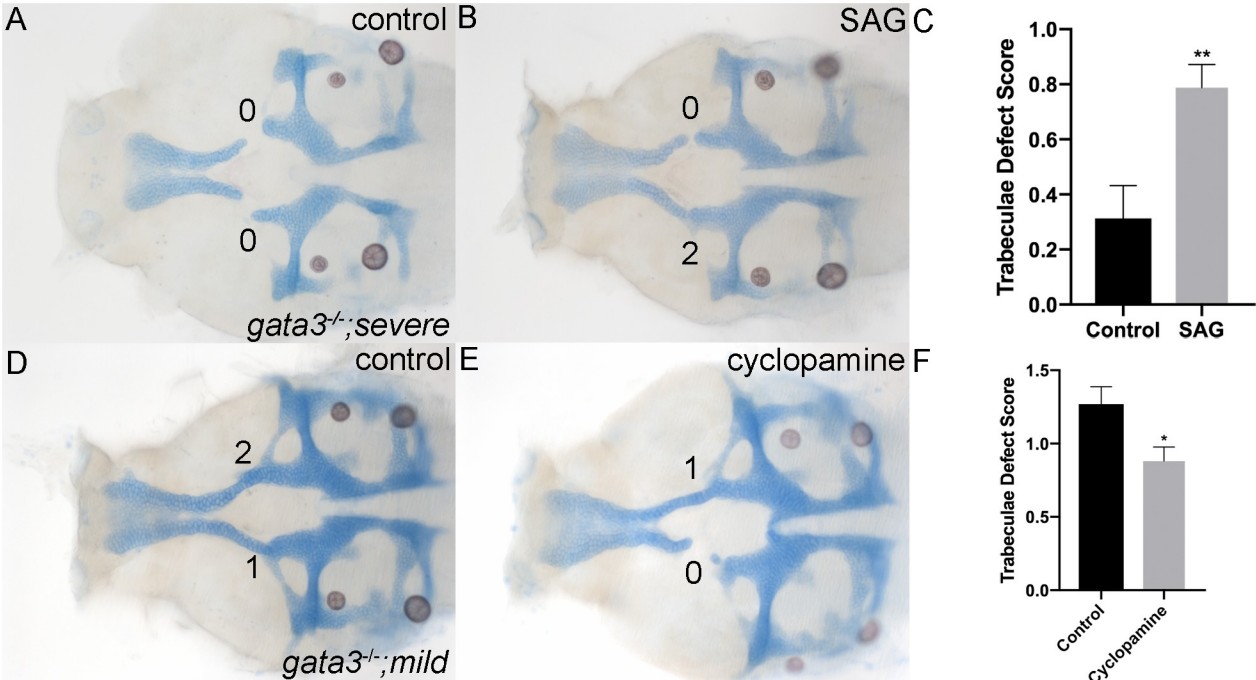

**Fig 8. Shh signaling can modulate the trabeculae phenotype in *gata3* mutants.** (A-B) Flat mounts of 5dpf *gata3 b1075 /b1075* "severe" background neurocrania anterior to the left. (A) Phenotype of "severe" *gata3 -/-* with characteristic severely affected trabeculae. (B) Representative image of a *gata3 -/-* "severe" embryo treated with the Hh agonist SAG, showing the partial rescue of the trabeculae. (C) Graph depicting trabeculae defect scores for untreated and SAG-treated *gata3 -/-*. The trabeculae phenotype was significantly improved in the SAG-treated embryos compared to controls via t-test (p = 0.0022). Control embryos x = 0.3125, SEM = 0.1197, n = 16; SAG treated embryos x = 0.7879, SEM = 0.08436, n = 33). (D-F) Flat mounts of 5dpf *gata3 b1075 /b1075* "mild" background neurocrania anterior to the left. (D) Phenotype of "mild" *gata3 -/-* showing mildly affected trabeculae. (E) Typical image of a *gata3 -/-* "mild" embryo treated with the Shh antagonist cyclopamine displaying a much more severe phenotype. (F) Graph depicting trabeculae defect scores for untreated and cyclopamine-treated *gata3 -/-*. Trabeculae phenotype was significantly exacerbated in the cyclopamine-treated embryos compared to controls as determined by t-test p = 0.0174, Control x = 1.269, SEM = 0.1184 n = 26; cyclopamine-treated embryos x = 0.88, SEM = 0.0975, n = 50).

disrupted stacking of chondrocytes in the trabeculae (S3 Fig). While this mutant also has characteristic midline defects, here we are interested in the trabeculae defects. This similar phenotype in the trabeculae suggests that the Shh pathway may be acting in concert with Gata3 to promote palatogenesis. We took advantage of our inbred *gata3b1075* mutant lines selected for severe and mild phenotypes to test the effects of modulating Shh signaling on *gata3* phenotypes.

If Shh promotes palatal development independent of Gata3 function, then elevation of Shh signaling should ameliorate the phenotypes of *gata3* mutants. We elevated Shh signaling using the Smoothened agonist SAG. We found that a treatment with 12.5 uM SAG did not alter wild type development but improved the phenotypes of *gata3* mutants in the severe background (Fig 8). While mutants in the severe genetic background typically lack trabeculae (Fig 8A and 8C), treatment with 12.5 uM SAG resulted in a significant improvement in the trabeculae score (Fig 8B and 8C). Thus, consistent with our model, elevating Shh signaling is able to partially compensate for the loss of *gata3*.

This model also predicts that reduction in Shh signaling will exacerbate the phenotypes in our mild *gata3* mutants. We used 12.5 uM cyclopamine to reduce Shh signaling to a level that doesn't disrupt trabeculae development in either of the genetic backgrounds used in these analyses (n = 26 WIK; n = 28, EK), we note that typical zebrafish doses to disrupt craniofacial development at this age range between 50 and 100 uM. We find that this low dose of

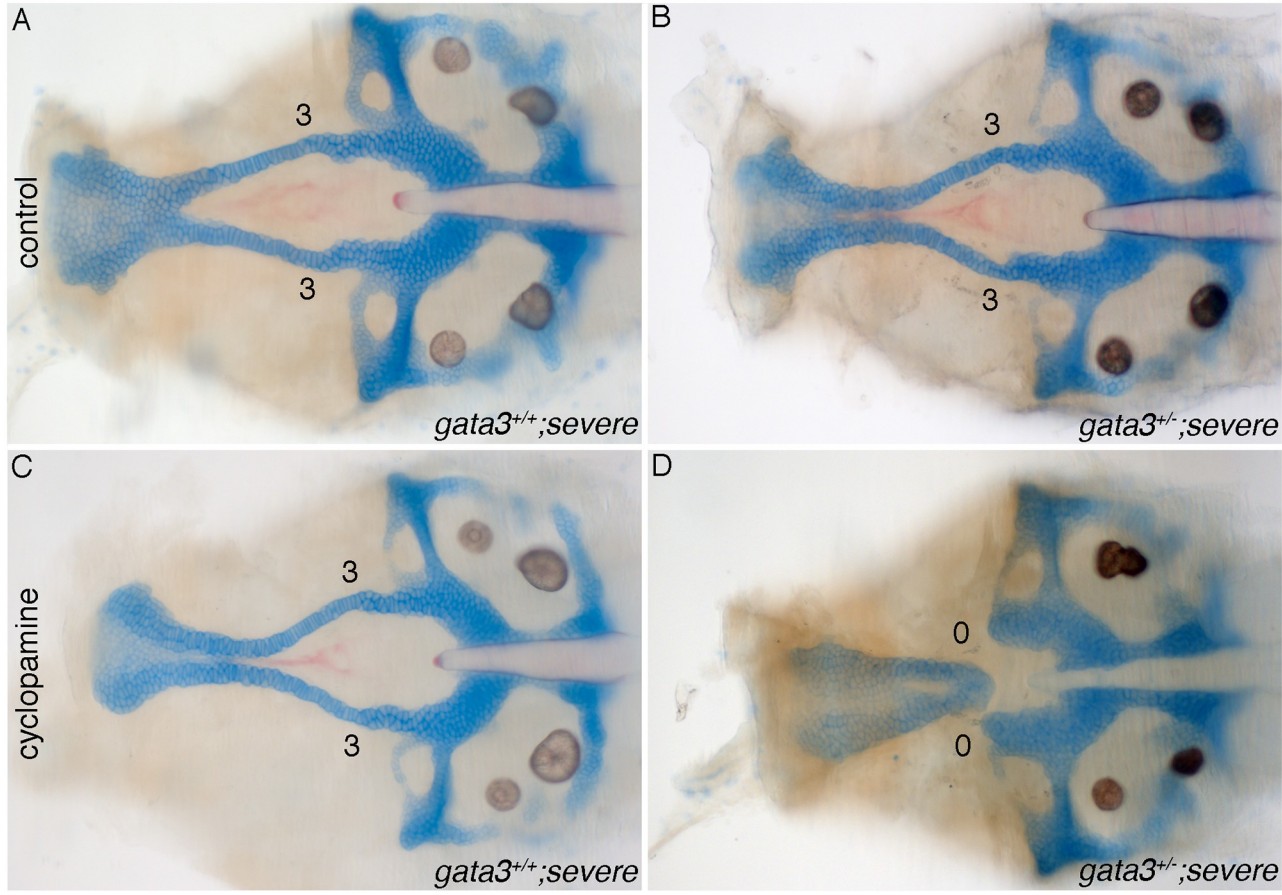

**Fig 9. Reduction of Shh signaling can induce *gata3* haploinsufficiency.** (A-D) Flat mounts of 5dpf *gata3* [b1075/b1075] neurocrania anterior to the left. (A-B) control untreated-embryos from the *gata3* "severe" background showing normal stacking and fusion of trabeculae. Both of (A) wild type and (B) heterozygotes develop normally. (C-D) Cyclopamine-treated embryos. (C) Wild type embryo showing normal stacking and fusion of trabeculae (n = 22/22). (D) Heterozygote embryo displaying complete loss of trabeculae with severe clefts (n = 6/61).

cyclopamine is capable of exacerbating the palatal phenotypes in mild *gata3* mutants relative to vehicle-treated mutants (Fig 8). Collectively, these findings demonstrate that the overall level of Shh signaling is capable of modifying, in both directions, the phenotypes of *gata3* mutants.

Many human diseases, including HDR, are due to haploinsufficiency. Based on our findings with Shh, we sought to determine if a low dose of cyclopamine could cause haploinsufficiency in the severe *gata3* mutant background. In vehicle-exposed embryos, we find that all *gata3* heterozygotes develop identical to their wild type siblings (Fig 9A and 9B). While 12.5 uM cyclopamine has no effect on wild type fish (Fig 9C), 10% of heterozygous fish had palatal defects, mirroring the haploinsufficiency observed in human HDR patients (Fig 9D, n = 6/61).

These findings suggest that there may be cross talk between Gata3 and Shh. Higher doses of SAG or cyclopamine do not alter the expression of *gata3* (S4 Fig). In order to determine if alterations in Shh signaling modulates the expressivity of *gata3* mutants, we turned to quantitative fluorescent *in situ* hybridization across *gata3* mutant backgrounds. We found that the levels of *ptch2* were equivalent in wild type embryos across the two genetic backgrounds (Fig 10). Interestingly, relative to the respective wild type embryos, "mild" mutants significantly upregulate the expression of *ptch2*, while "severe" mutants do not (Fig 10, see S5 Fig for

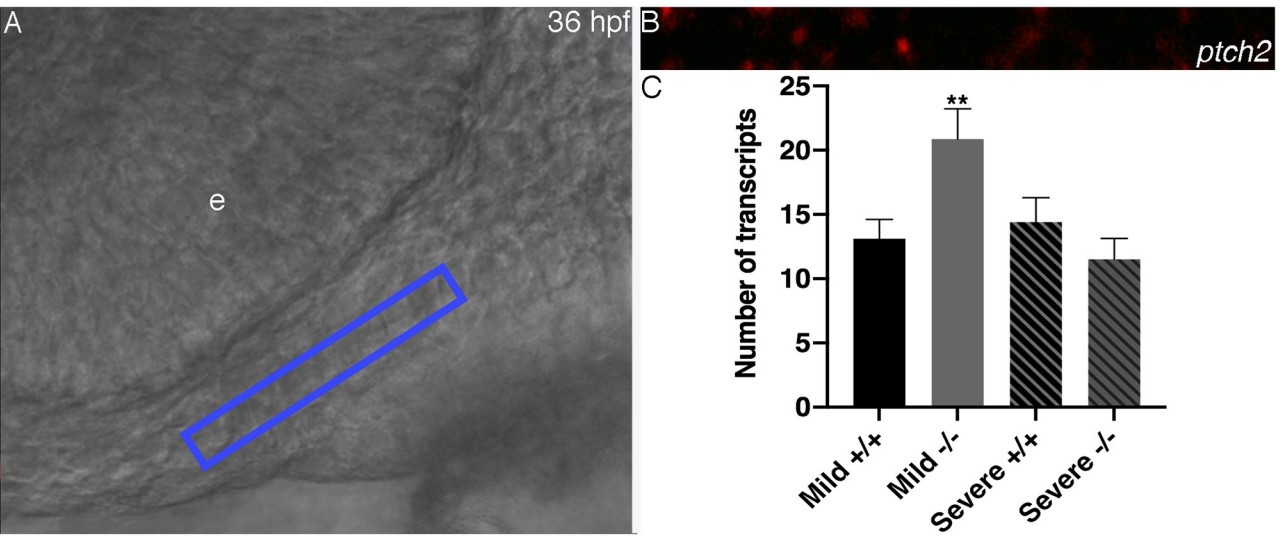

**Fig 10. Upregulation of Shh signaling in mild mutants.** (A) Lateral view of 36 hpf embryo, blue box indicates region of interest in the maxillary trabeculae precursors. (B) Confocal z slice of region of interest in (A). Red dots are labeled *ptch2* transcripts. (C) Quantification for each group. Mild mutants have significantly elevated levels of *ptch2* relative to the severe mutants (p = 0.0085) and the mild wild types (p = 0.0183). Mild wild type x = 13.11, S.E. = 1.506, n = 18; mild mutant x = 20.87, S.E. = 2.360, n = 15; severe wild type x = 14.40, S.E. = 1.904, n = 15; severe mutant x = 11.50, S.E. = 1.640, n = 12.

zoomed out views). Collectively, these findings strongly suggest that a compensatory upregulation of Hh signaling can lessen the phenotypic outcome of loss of Gata3 function.

## Discussion

### Gata3 functions downstream of Bmp signaling in neural crest precursors of the palatal skeleton

Despite its known involvement in palatal development, the role of Gata3 in this process is uncharacterized. We have found that *gata3* is required for proper development of a region of the zebrafish palatal skeleton, the trabeculae. The expression of *gata3* localizes to regions that contribute to the trabeculae [2,3,6]. Consistent with its expression pattern, we find that neural crest cells require Gata3 function for palatogenesis. While detailed expression analyses during palatal development in other species have not been performed, *Gata3* is strongly expressed in maxillary/frontonasal regions of mouse and chicken embryos [29,30,36]. Thus, Gata3 is likely to have evolutionarily conserved functions in palatal development. Analyses in conditional mouse mutants would aid in characterization of these conserved functions.

The regulation of *Gata3* expression also appears evolutionarily conserved. Using *BRE: d2GFP* transgenics, we demonstrate that the neural crest cells that express *gata3* receive Bmp signaling. The expression of *gata3* in the neural crest, but not other tissues, is lost when Bmp signaling is reduced in hypomorphic *smad5* mutants, demonstrating that Bmp signaling is necessary for *gata3* expression. While their analyses focused on mandibular development, Bonilla-Claudio and colleagues demonstrate that over action of Bmp results in the expansion of *Gata3* expression in the frontonasal and maxillary prominences [30]. Interestingly, the Drosophila homologue of Bmp (*decapentaplegic)* induces Gata (*pannier*) expression during development of the notum [37]. Thus, Bmp-Gata signaling may have arisen very early in animal evolution.

It is likely that Bmp signaling directly activates *Gata3* expression. The time windows when forced expression of human *GATA3* can partially rescue *gata3* mutants and DN-Bmpr1a transgenics are largely overlapping. Work in the mouse mandible has demonstrated that Smad1/5 binds a region upstream of *Gata3* [30]. While these studies did not assess Smad1/5 binding in maxillary neural crest, these collective findings prompt similar analyses in maxillary crest.

## Interactions between Bmp and Shh signaling modulate variability

Three general mechanisms could explain the variation in *gata3* mutant phenotypes: 1) Residual gene function in the mutant, 2) partial compensation via other Bmp target genes and 3) phenotypic modulation via a different signaling pathway.

**Residual gene function.**    Previously, we demonstrated that a point mutation of *gata3* generated phenotypes that were highly variable dependent upon genetic background [13]. This *b1075* allele mutated a Cysteine that is homologous to one disrupted in HDR syndrome and required for coordinating the second zinc finger necessary for DNA binding [13,32]. The human *GATA3* mutation (Cys318Arg) was hypothesized to be a null based on *in vitro* data [32]. However, the variability in *b1075* mutants left open the possibility that these missense mutants retained some functionality *in vivo*.

Our results here strongly support the model in which the human Cys318Arg mutation is a null. The *au42* allele generates the same range of phenotypes as does *b1075*, although we have not generated inbred lines within different genetic backgrounds to test for modulation by genetic background. The *au42* mRNA encodes a protein predicted to terminate before the zinc finger domains, which are necessary for Gata3 function. Thus, the variability in *gata3* mutant phenotypes is unlikely to be due to residual gene function.

**Partial compensation.**    Most major signaling pathways have a multitude of target genes that behave in networks to drive development. The Bmp pathway is no exception with a growing number of characterized targets in craniofacial development generally [30,38,39] and palatogenesis more specifically [4]. One or more of these targets could be partially redundant with *gata3*. For instance, both *Gata3* and *Gata2* are regulated by Bmp during mandibular development [30] and these same two genes are partially redundant during trophoblast development [40]. This model predicts that attenuation of Bmp signaling in *gata3* loss-of-function embryos will worsen the resulting phenotypes. However, we did not detect such an interaction. While we cannot rule out potential redundancy between Bmp targets in the generation of the trabeculae phenotypes, it would appear that this is not the major cause of phenotypic variability in *gata3* mutants.

**Interaction between pathways.**    A wide array of signaling pathways are involved in palatogenesis, with the Bmp, Fgf and Shh pathways being particularly important [4]. Cross talk between the Bmp and Shh pathways has been demonstrated in palate development [41]. In addition to their roles in palatogenesis, interaction between the Bmp and Shh pathways appear to mediate facial shape [42]. We and others have shown that both of these two pathways are crucial for palatogenesis in zebrafish [2,3,6,35]. Interestingly, Gata6 has been shown to bind Gli transcription factors [33]. Cross-talk between these pathways provides mechanisms for interactions to generate variability. Our findings do not rule out the involvement of other pathways in modulating the phenotypes in *gata3* mutants. Indeed, we have previously demonstrated that altering Hsp90 activity can change the phenotypes in these mutants [13]. While the mechanism of this interaction is unknown, Hsp90 has a large number of client proteins involved in signaling [43]. However, our results clearly demonstrate that interactions between Shh and the Bmp-Gata3 pathway modify the phenotypes generated in *gata3* mutants.

### Implications for variation in human birth defects

Our current findings implicate a Bmp-Gata3 signaling pathway in the genesis of microsomia and HDR-related craniofacial defects (including palatal defects), both highly variable craniofacial defects. There are no known causative variants for craniofacial microsomia in humans. However, *GATA3* is associated with microsomia, a disease that is thought to be caused by gene-environment interactions [23–25]. Our results provide strong evidence that *GATA3* is responsible for craniofacial microsomia and likewise implicate Bmp pathway members in the etiology of microsomia.

Phenotypic variation is a common theme in birth defects. Such variation is often attributed to genetic background and gene-environment interactions. Our results demonstrate that both of these processes can alter the phenotypes of *gata3* mutant. We previously demonstrated that genetic background strongly modified the phenotype resulting from *gata3* mutation [13]. Here we demonstrate that this difference in expressivity is modulated via the level of Shh signaling. Interestingly, we find that *gata3* mutants in the "mild" genetic background have a compensatory upregulation in Shh signaling. Our inhibitor and agonist studies demonstrate that altering Shh signaling modifies the phenotypic outcomes in *gata3* mutants. Therefore, the upregulation of Shh signaling in the "mild" background is likely to be an important contributor to the less severe phenotype. The precise nature of the compensatory upregulation of Shh signaling is unknown, but will provide important insight into developmental robustness.

Our finding that Shh signaling can modulate the phenotypes in *gata3* mutants has important implications for understanding the potential for gene-environment interactions in birth defects. Haploinsufficiency for Shh signaling pathway members underlie holoprosencephaly, another disease that is thought to be caused by gene-environment interactions [44–46]. A growing body of research demonstrates that the Shh pathway is extremely sensitive to environmental attenuation. For instance, the common chemical synergist piperonyl butoxide (PBO) and many dietary molecules such as tomatidine and solanidine can inhibit Shh signaling [47–49]. Microsomia is thought to have a large environmental component [26]. The interaction between cyclopamine and Gata3 implicate similar influences, with environmentally relevant molecules such as PBO in the genesis of microsomia and HDR syndrome. Collectively, these results suggest that alterations to Gata3 and its associated regulatory network destabilizes craniofacial development and that the overall level of Shh signaling modifies the response to this destabilization.

## Materials and methods

### Ethics statement

**For the animal experiments, all procedures were performed according to an approved IACUC protocol** (AUP00002018) at UT Austin.

### Fish care and fish lines

Zebrafish (*Danio rerio*) were raised according to [50,51] and were staged as previously described [52]. The following transgenic lines were used: *BmpRE-AAV.mlp:d2GFP^{mw30}* [53], *hsp70I:dnXla.BmpR1a-GFP^{w30}* [54], *Tg(fli1:EGFP)^{v1}*[55], *Tg(sox10:mRFP)^{vu2}* [56], *Tg(3.5ubb: LOXP-EGFP-LOXP-mCherry)*[57], *Tg(hsp70l:Gal4)* [58] and are referred to as *BRE:d2GFP*, *hsp:DN-BmpR1a*, *fli1:EGFP*, *sox10:mRFP*, *ubi:Switch* and *hsp:Gal4* for clarity. The *smad5^{b1100}* allele is described [6]. The *gata3^{b1075}* allele is described; the "mild" background was generated by outcrossing to WIK and the "severe" background was generated in a cross to *fli1:EGFP* (in the EK background)[13]. Wild type stocks were AB unless otherwise noted.

## Generation and genotyping of *gata3*[au42] CRISPR line

We utilized ZiFiT Targeter (http://zifit.partners.org/ZiFiT/) to identify gRNA binding sites for *gata3*. With these target sites, we made *gata3* gRNA via MEGAscript T7 Kit and Cas9 mRNA via T3 mMessage Kit (Invitrogen) using a described protocol [59]. Embryos were injected with a 2nl bolus of a cocktail containing: 100ng/ul *Cas9* mRNA, 50ng/ul *gata3* gRNA in water and phenol red (to visualize the injection).

Embryos were genotyped using either Restriction fragment length polymorphism (RFLP) or High-resolution melt (HRM) analysis. For RFLP we used forward 5'-GGTATGACGAAT CCCACAACAGAC-3' and reverse 5'- AAGAGGACCCACCTATCAGGCTAC3' primers. Digestion with NlaIV results in a 531bp mutant band and 394 and 139 wild type fragments. Primers for HRM analyses were forward GGCAAATCTATCGGCCCTCA and reverse GGACAGCGAGGAGGAAGAAG. The resulting product is 129 bp.

## Generation of *Tg(hsp70:GATA3-EGFP)*[au34]

We obtained full-length human *GATA3* in pDonor from human ORFeome Version1.1 (cat# OHS1770-9380128, genbank access #CV025706). Using the Tol2 kit [60], we recombined the *GATA3* MEC vector, the P5E-hsp70 (#222, 5' entry clone), P3E-EGFP (#366, 3' clone) and destination vector pDest cy2 (#395) to make a 10267 bp vector. A mix of 100 ng plasmid, 155 ng transposase RNA and phenol red were injected as a 3nl bolus into AB fish and screened for glowing hearts. We refer to the recovered transgenic line as *hsp:GATA3-EGFP* for clarity.

## Tissue labeling and imaging and analyses

Cartilage and bone staining was done as previously described [61]. A modification of a published protocol [52] was used to visualize the dissected neurocranium. In particular, the brain and overlying ectoderm was left intact to hold cartilages and bones in their context. The *gata3* riboprobe is described [62]. RNA *in situ* hybridization was performed according to published protocols [63]. Fish used for in situ analyses were raised in 0.0015% PTU (1-phenyl 2-thiourea) to inhibit melanin production. Activation of all heat shock lines was performed by placement in a 39°C water bath for 50 minutes. Fluorescent whole mount *in situs* were performed using RNA Scope version 2. ACD Biology designed probes to *gata3*, *ptch2* and *pdgfra*. We modified the existing version 2 protocol and the whole mount zebrafish technical note outlined by ACD Biology. We also integrated published recommendations [64]. A complete protocol is outlined (S6 Fig).

Colorometric *in situ* hybridizations and cartilage/bone stained embryos were imaged on a Zeiss Axioimager. Confocal images of embryos were collected on a Zeiss 710 microscope. All images were processed in Photoshop CS3. All statistics and graphs were generated in Prism versions 7.0–9.0.

For quantification of *ptch2* expression, we used identical imaging settings for all embryos. We used a 40x water lens with a 2x digital zoom. For each embryo three individual z sections (4 um apart, a distance which prevented the same dot from being sampled across two z sections) were selected for counting dots of labeled *ptch2* probe. For consistency, we selected a 5 um x 50 um region of maxillary neural crest cells immediately adjacent to the oral ectoderm and below the eye, known to contribute to the anterior neurocranium, for counting. The same region was selected across embryos using transmitted light initially and was verified as neural crest cells using *pdgfra* as a neural crest cell marker. The number of dots was manually counted. All counts were performed blinded to genotype.

## Chimera analyses

We generated two types of genetic chimeras by neural crest transplantation from *sox10:mRFP* into *gata3* [au42];*fli1:EGFP* embryos and *ubi:Switch* into *gata3 LOF;sox10mcherry* as described [65]. For *gata3* LOF embryos we injected 15ng of a *gata3* morpholino as described [13]. The resulting embryos were imaged at 30 hours post fertilization (hpf) to access contribution of donor cells to the maxillary region of the first arch. To characterize *ubi:Switch* neural crest cell incorporation into the neurocranium the viserocranium was dissected away and the neurocranium was imaged at 4dpf. Neurocrania were then stained with Alcian Blue and Alizarin Red.

## Chemical treatments

Cyclopamine and SAG treatments were performed as described elsewhere [66,67]. Both cyclopamine and SAG were used at 12.5uM, a dose that did not disrupt development in wild type embryos. Embryos were treated from 24 to 30 hpf. Embryos were grown to 5 dpf and stained for cartilage and bone.

## Supporting information

**S1 Fig. Cartilage and bone stained 5 dpf larvae.** (A-D) Representative embryos of each genotype were imaged laterally with head to the left. Blue and red staining indicate cartilage and bone respectively (A) non-heat shocked wild type zebrafish. (B) Heat shocked wild type fish have normal morphology. (C) Heat shocked *hsp:DN-Bmpr1a* fish show some disruption to normal development, cardiac edema and alterations to the face. (D) Heat shocked *hsp: DN-BmpR1a;hsp:GATA3-GFP* fish appear normal. (E) Heat shocked *hsp:DN-BmpR1a;hsp:Gal4* show similar disruptions to development as in C, providing genetic evidence that a double heat-shock transgenic does not interfer with transgenic expression off of the heat shock promoter. (F) Overall, *gata3* mutants have normal morphology. The image is of a *gata3*[au42] mutant. (TIF)

**S2 Fig. Double heat shock line does not diminish EGFP signal.** (A-B) Confocal single-z images of embryos at 30 hpf, embryos were heat shocked at 24 hpf. Representative images of (A) a *hsp:DN-BmpR1a* embryo and (B) a *hsp:DN-BmpR1a;hsp:GATA3-EGFP* embryo, arrows indicate cell membrane expression of DN-BmpR1-EGFP. Intensity values for *hsp:DN-BmpR1a* (n = 11, mean = 8663 arbitrary units (should be added for clarity), standard deviation = 1585, n = 11) were not statistically different from the intensity values for *hsp:DN-BmpR1a;hsp: GATA3-EGFP* (mean = 10583, standard deviation = 3463, n = 11), p-value = 0.1101 by t-test. (TIF)

**S3 Fig. Trabeculae defects in a *shh* hypomorph.** Flat mount of a 5dpf *shh*[tq252/tq252] mutant neurocranium anterior to the left. Arrow is pointing to disrupted stacking of trabeculae cells. (TIF)

**S4 Fig. Shh does not affect the expression of *gata3*.** (A-D) Lateral views of embryos at 36 hpf showing normal expression of *gata3* regardless of treatment. (A) Embryo treated with DMSO as a SAG treatment control. (B) Cyclopamine treatment control embryo treated with vehicle ETOH. (D) Embryo treated with 25*uM* SAG (C). 25 uM cyclopamine treated embryo. (TIF)

**S5 Fig. First arch images of *ptch2* expression.** Single-z images of DIC and patched expression in the first pharyngeal arch. Arrows indicate maxillary neural crest. Eye is indicated by e. (TIF)

**S6 Fig. Whole mount RNA Scope version 2 fluorescent in situ protocol.**
(DOCX)

## Acknowledgments

We thank the members of the Eberhart lab for manuscript and figure design suggestions. We thank Angela Martinez for excellent fish care. We thank Thomas Schilling for the *hsp:DN-BmpR1a* line and Brian Link for the *BRE:d2GFP* line.

## Author Contributions

**Conceptualization:** Mary E. Swartz, Johann K. Eberhart.

**Data curation:** Mary E. Swartz.

**Formal analysis:** Mary E. Swartz.

**Funding acquisition:** Johann K. Eberhart.

**Investigation:** Mary E. Swartz, C. Ben Lovely.

**Methodology:** Mary E. Swartz, Johann K. Eberhart.

**Project administration:** Johann K. Eberhart.

**Resources:** Johann K. Eberhart.

**Supervision:** Johann K. Eberhart.

**Validation:** Mary E. Swartz.

**Visualization:** Mary E. Swartz.

**Writing – original draft:** Mary E. Swartz.

**Writing – review & editing:** Mary E. Swartz, C. Ben Lovely, Johann K. Eberhart.

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
