## [Decision Letter · Decision Letter 0]

29 Jul 2020

Dear Dr Swartz,

Thank you very much for submitting your Research Article entitled 'Severity of craniofacial disease phenotypes caused by Gata3 loss of function are modulated by parallel Shh and Bmp signals' to PLOS Genetics.

The manuscript was fully evaluated at the editorial level and by three peer reviewers. As you will see, the reviewers express mixed enthusiasm for the work, with the most serious concerns raised by reviewer #3 regarding strength of advance. The manuscripts and the reviews have now been discussed among members of the editorial board. Overall, we are interested in the question and the approach, but agree with the concerns raised by all reviewers with regard to rigor, interpretation, and presentation. 

We are willing to consider a major revision, but caution that success of a revised manuscript will require carrying out the additional work noted by reviewers #1 and #2, and addressing the concerns from reviewer #3 in the "reply to reviewers" and/or changes to the presentation.* *Should you decide to revise the manuscript for further consideration here, your revisions should address the specific points made by each reviewer. We will also require a detailed list of your responses to the review comments and a description of the changes you have made in the manuscript.

If you decide to revise the manuscript for further consideration at PLOS Genetics, please aim to resubmit within the next 60 days, unless it will take extra time to address the concerns of the reviewers, in which case we would appreciate an expected resubmission date by email to plosgenetics@plos.org.

We are sorry that we cannot be more positive about your manuscript at this stage. Please do not hesitate to contact us if you have any concerns or questions.

Yours sincerely,

Robert A. Cornell

Guest Editor

PLOS Genetics

Gregory Barsh

Editor-in-Chief

PLOS Genetics

Reviewer's Responses to Questions

**Comments to the Authors:**

Reviewer #1: This study provides new evidence that Gata3 function in the early postmigratory neural crest mesenchyme is required for palatal skeletal formation in zebrafish. It also confirms that gata3 is a target of Bmp signaling in palatal precursors of the maxillary domain, as previously demonstrated for the mandibular domain. The most interesting aspect of this study is investigation into the variability of the gata3 phenotype. Through construction of a new early-stop allele, they show that variability is not due to the original allele being hypomorphic. They then go on to show that modulation of Shh signaling can affect the severity of the gata3 palatal phenotype. This is an important result given the known variability of craniofacial defects upon altering Hh signaling in general and helps explain why mutations in other genes may result in variability of craniofacial defects. I think in general this is a well performed study that extends our knowledge of Gata3 function in the palate. My main criticism is that, while they certainly show that experimentally modulating Shh signaling can alter the phenotype, they do not show that stochastic changes in Shh signaling are the basis of variability in gata3 mutants. Modulation of critical craniofacial pathways, including Shh but also Hsp90 (as published earlier by the authors) and likely many other genes/pathways, is expected to enhance the phenotype, but the authors provide no direct data that Shh signaling is lower in severe versus mild strains of the gata3 mutant. If the authors can provide new data on Shh signaling in severe vs. mild strains or tone down their claims that Shh is the cause of variability (in addition to addressing some other moderate concerns), then I think this will be a strong study of interest.

1. The description of parallel pathways is confusing in several places. For example, the Title could be interpreted that Bmp works in parallel to Gata3 which is not what the authors likely intend (language is unclear whether Bmp would be in parallel to Shh, or whether Bmp and Shh are in parallel to Gata3). Lines 48-49 (and again on lines 55-56) have similar problematic language: “However, Bmp attenuation does not alter phenotypic variability in gata3 loss-of-function embryos, implicating a parallel pathway.” I think the authors mean to use this to introduce Shh, but it could wrongly be interpreted as Bmp being a parallel pathway to Gata3.

2. They do not demonstrate stochastic differences in Shh signaling, just that changing Shh experimentally can alter severity. Rather than stating that their data support Shh differences being the cause of variability, they should more accurately state that their experimental data raise the possibility that Shh could be one candidate contributing to variability. For example the statement on Lines 68-70 is too strong: “Instead, we identify a separate signaling pathway,the Sonic Hedgehog (Shh), pathway that is responsible for the variability in gata3 mutant defects.” The authors previously showed in Sheehan-Rooney, 2013 (which should be cited and discussed) that modulation of chaperones Ahsa1 and Hsp90 similarly alter gata3 mutant spectrum, showing that the Shh effect described here is not unique. It may very well be that alterations of many genes/pathways can alter the gata3 phenotype – this is a standard synergistic/additive effect in genetics, i.e. genetic enhancers/suppressors. In order to further support a direct effect of Shh in variability, the authors would have to show different levels of Shh signaling in mild versus severe gata3 strains (e.g. by quantitative gli1, ptc in situs). The Introduction and Discussion should also be revised to imply that this study addresses stochastic effects, as opposed to the genetic interactions it does address.

3. In Fig. 1C, the authors show high NMD in the au42 allele. They need to address the possibility of genetic compensation as described by the Stainier lab and others, for example in the Discussion where they address partial compensation.

4. Nice result that only maxillary gata3 expression reduced in smad5 mutants. Would be nice to test directly if DN-Bmp reduces gata3 as this is genetic tool used to probe functional consequence of reduced Bmp signaling. Injecting gata3-MO into smad5 hets does not test for parallel vs. epistasis as smad5 is just sensitized background at best. It would be better to test effect of injecting gata3-MO into dn-Bmp fish and whether any enhancement occurs. Representative images should also be shown as part of a main figure as this is important result.

5. To truly test whether Shh functions independent of gata3 regulation, in situs should be performed in SAG and cyclopamine fish to analyze effects on gata3 maxillary expression (as done for smad5 mutants).

6. In Fig. 3A, what is evidence that gata3 at 22hpf is in endoderm and not mesoderm or neural crest of posterior arches? Similarly at later stages, how is neural crest staining in the maxillary domain confirmed? Ideally, double in situs would show gata3 expression in neural crest mesenchyme.

Minor comments:

-In Fig. 2, the difference between phenotypic score 2 and 1 is not obvious.

-In Fig. 2, inappropriate lateral commissure fusion should be indicated with arrows.

-Fig. 6 and 7 should be combined as they both address Bmp signaling.

-Fig. 9,10,11 should be combined as they all present similar analysis of Shh manipulation. Fig. S1 should also be incorporated into this main figure. The stacking defect of shha hypomorphs is of unclear relevance to the gata3 phenotype as it has a midline cartilage bridge not apparent in any of the images of the gata3 mutants. This should be better explained or perhaps removed from manuscript.

-In Fig. 8D, use of au43 is confusing as this allele not stated elsewhere in figure and could be confused with au42 lof allele. Would be better to state "DN; GATA3-GFP"

Reviewer #2: Here the authors describe how BMP and Shh signaling interact with genetic disruption of gata3. They found that BMP is likely upstream of gata3 and required for gata3 expression. They also show that chemically altering Shh signaling can modify the gata3-null trabeculae phenotype. Overall, I find this to be an important and well-done study of phenotype variability in craniofacial development. However several control experiments need to be added and there is a concern with sample size differences between experimental treatments/controls. Otherwise this work is well done and warrants publication with these minor revisions.

Major Comments:

1) In Figure 1 the RT-PCR needs to include an endogenous control to show equal mRNA levels. Also, why are the WT and au42 bands so different in size?

2) Figure 2 (and in general), is there an interaction between severity of defect on contralateral sides? Does a 0 on one side predispose to a defect on the other side, etc.? Also, in Figure 2 are the gata3-/- embryos also stained for alizarin, as is the wildtype?

3) In Table 1, there are large disparities in the sample sizes between different times of treatment. This makes interpretation of the data challenging. For instance, is treatment at 18hpf not significant because only 12 embryos were examined? Whereas at 24hpf 73 were examined?

4) Regarding the transplant experiment in Fig. 5: if there is variability in the trabeculae phenotype on the right versus left side (as stated by the authors) then using the contralateral/non-transplanted side is not an appropriate control. It is possible that “rescuing/altering” one side with gata3 WT cells affects the phenotype of the contralateral side. The authors need to perform additional experiments where gata3-/- cells are transplanted into gata3-/- hosts and transplanted vs. non-transplanted sides are compared.

5) WISH staining can’t really be quantified, so quantitative PCR of gata3 level could be useful.

6) Regarding cyclopamine treatment in Fig. 11: there were many fewer wildtype embryos treated with cyclopamine than gata3 hets (22 vs. 66). Only 6 gata3 hets showed a defect. We do not know that treating 66 wildtype embryos would result in a similar number of defects. This makes interpreting the data difficult. Also, for completeness, phenotypic quantification should be presented as was done for other experiments.

Minor comments:

1) Referring to the anterior neurocranium of the zebrafish as the “palate” is not agreed upon in the field. The embryology and ontogeny of this ethmoid plate structure has been described by many authors that are cited. While I personally totally understand why this term “palate” is used here, one may be more precise to just reference the structure as anterior neurocranium on a consistent basis.

2) Further, it is ill advised to refer to lack of trabeculae fusion to the posterior neurocranium as a “cleft” or “clefting,” especially since the embryologic origin of the posterior neurocranium is not from the second stream of hox-free cranial NCC that contributes to the maxillary-derived structures. BTW, the correct clinical term is “cleft” and not “clefting,” despite this incorrect usage having promulgated throughout the craniofacial literature.

3) There are numerous typos and punctuation mistakes. Also “wildtype”, “wild type” and “wild-type” were all used in the manuscript. This should be made consistent.

Reviewer #3: This paper addresses the causes of phenotypic variability in GATA3 mutants that cause human HDR syndrome, microsomia and choanal atresia, using zebrafish as a model. Phenotypic variability is an important problem and its causes are difficult to resolve. Two fish Gata3 mutants show variable defects in the presumed fish equivalent of the primary “palate”. This appears to reflect a requirement in embryonic neural crest cells, and a combination of genetic and pharmacological experiments suggest that this variability reflects fluctuations in Bmp and Shh signaling.

Conditional overexpression experiments using heat shock drive GATA3 expression everywhere. Therefore any rescue of the trabeculae could reflect rescue of growth or patterning of other tissues, including the non-neural ectoderm, which is known to express Gata3 in zebrafish, and interacts with developing palate-forming neural crest.

A related concern is that the skeletal phenotypes in the trabeculae are subtle and potentially influenced by overall embryo health and growth. The skeletal preparations are very nice, but I would like to see the whole embryos in at least some of the rescue experiments, such as Hsp:GATA3-EGFP;Hsp:DN-bmpr1a double transgenics. Are these specific rescues of the skull or of the whole animal or whole head?

Many of the results are incremental advances beyond what is known. Requirements for Gata3 in neural crest and craniofacial development are known. Gata3 has been shown to be downstream of Bmp signaling and a direct transcriptional target of Smads in the mandible. Reductions in maxillary Gata3 expression in Smad5 mutants in Fig. 6, as well as expression of the Bmp reporter in maxillary precursors in Fig. 7 are clear but not novel.

Several interpretations are based upon negative results. Neither Gata3 mutant completely eliminates Gata3 mRNA, but the potential effects of any residual protein on phenotype or variability are discounted. Later it is mentioned that Gata3 knockdowns in Smad5 heterozygotes had no effect, also a negative result that is overinterpreted (p. 13).

Figures are minimal. Figure 1 could be supplementary. Many of the figures need better labels to make it easier for readers unfamiliar with the zebrafish skull.

Figure 5 is another minimalist figure and the result is unclear. Do the transplanted cells contribute to the rescued skeleton or not?

I do not understand the interpretation of the experiment with low dose treatments with cyclopamine of Gata3 mutants of different severity in Fig. 11. How does this show in any definitive way a role for Shh in variability?

The authors have not considered other possible signals that might influence variability in the Gata3 mutant phenotypes and this is not discussed.

**Have all data underlying the figures and results presented in the manuscript been provided?**

Reviewer #1: Yes

Reviewer #2: Yes

Reviewer #3: Yes

PLOS authors have the option to publish the peer review history of their article (what does this mean?). If published, this will include your full peer review and any attached files.

Reviewer #1: **Yes: **Gage Crump

Reviewer #2: No

Reviewer #3: No

---

## [Decision Letter · Decision Letter 1]

23 Feb 2021

Dear Dr Swartz

Thank you very much for submitting your Research Article entitled 'Variation in phenotypes from a Bmp-Gata3 genetic pathway is modulated by Shh signaling' to PLOS Genetics.

The manuscript was fully evaluated at the editorial level and by independent peer reviewers. The two reviewers of the revised manuscript appreciated the attention to an important problem, and acknowledged the difficulty of  conducting experiments during the pandemic. However, each raised a distinct and substantial concern about the current manuscript that led them to question its conclusions of a neural-crest specific function for Gata3.

Based on the reviews, we will not be able to accept this version of the manuscript, but we would be willing to review a much-revised version. Specifically, an explanation for how *gata3* over-expression rescues phenotypes in DN-BMPR embryos outside of the maxillary domain where *gata3* is expressed must be provided or the reviewer's suggestion to remove the result should be adopted. In addition, transplant experiments where donor-derived cells beyond just the neural crest are visualized are necessary. Finally, the variablity in posterior trabeculae due to fish health should be explicitly addressed. We cannot, of course, promise publication at that time.

If you decide to revise the manuscript for further consideration at PLOS Genetics, please aim to resubmit within the next 60 days, unless it will take extra time to address the concerns of the reviewers, in which case we would appreciate an expected resubmission date by email to plosgenetics@plos.org.

[LINK]

We are sorry that we cannot be more positive about your manuscript at this stage. Please do not hesitate to contact us if you have any concerns or questions.

Yours sincerely,

Robert A. Cornell

Guest Editor

PLOS Genetics

Gregory Barsh

Editor-in-Chief

PLOS Genetics

Reviewer's Responses to Questions

**Comments to the Authors:**

Reviewer #1: The authors have largely addressed my initial concerns, except for one major concern and one more minor in nature.

Major: In Fig. S1, why does hsp:GATA3 also rescue the small eye defect and cardiac edema of hsp:DN-Bmpr1 mutants? There is a concern that rescue is possibly due to a non-specific effect of hsp promoter squelching. It is also very surprising that GATA3 can rescue all the defects of Bmp reduction as in different parts of the body presumably Bmp would have multiple targets. In other words, having two hsp transgenes may lead to "rescue" simply be causing reduced activity of the hsp:DN-Bmpr1a transgene due to competition with factors that bind to hsp promoters. Unless the authors have a good way of addressing this caveat (I can't think of one), this finding should be removed from the paper rather than just qualified with a comment such as "although we can't rule out....".

Minor: The new data showing upregulation of ptch2 in mild gata3 mutants are compelling. While I understand the variability in expression and the need to carefully quantitate expression in just the maxillary regions, images of ptch2 in each condition for the whole head (lateral views) should be shown for perspective. For example, is the ptch2 upregulation in mild gata3 mutants just restricted to the maxillary domain?

Reviewer #3: The revisions have addressed several of my comments and those of other reviewers, improving the paper, though the responses selectively leave out some points. My major remaining concern surrounds the rescue experiments and transplants, since both may rescue surrounding tissues. The transplants shown are not controls for cell type specificity since they only show sox10:mCherry positive donor NC cells, and not any other sox10-negative cells that were very likely included in the transplants. They also do not include evidence of contributions of transplanted cells to the skeleton on the rescued side (as requested in the initial review and not mentioned in the response). I realize this is difficult to address in with current restrictions, but it is pretty central to the interpretation. I also still worry that subtle phenotypes affecting posterior trabeculae are extremely variable depending not only on gene activity but fish health.

**Have all data underlying the figures and results presented in the manuscript been provided?**

Reviewer #1: Yes

Reviewer #3: Yes

PLOS authors have the option to publish the peer review history of their article (what does this mean?). If published, this will include your full peer review and any attached files.

Reviewer #1: No

Reviewer #3: No

---

## [Decision Letter · Decision Letter 2]

4 May 2021

Dear Dr Swartz,

We are pleased to inform you that your manuscript entitled "Variation in phenotypes from a Bmp-Gata3 genetic pathway is modulated by Shh signaling" has been editorially accepted for publication in PLOS Genetics. Congratulations!

Yours sincerely,

Robert A. Cornell

Guest Editor

PLOS Genetics

Gregory Barsh

Editor-in-Chief

PLOS Genetics

Comments from the reviewers (if applicable):

The reviewers find the current version substantially improved of the previous one. The editor agrees with a reviewer's suggestion to add labeling to Fig. S5.

Reviewer's Responses to Questions

**Comments to the Authors:**

Reviewer #1: The authors have now addressed all my concerns. The hsp:Gal4 and in particular hsp:DN-Bmpr1a-GFP imaging controls go a long way in addressing my doubts regarding the hsp:Gata3 rescue. I appreciate the authors taking my concern seriously and thoroughly ruling out the possibility of hsp promoter squelching. For the zoomed out images of ptch2 in the new Fig. S5, it may be helpful to outline or add arrows to designate the maxillary domain as this structure may not be apparent to non-experts.

Reviewer #3: I have no further concerns.

**Have all data underlying the figures and results presented in the manuscript been provided?**

Reviewer #1: Yes

Reviewer #3: Yes

PLOS authors have the option to publish the peer review history of their article (what does this mean?). If published, this will include your full peer review and any attached files.

Reviewer #1: No

Reviewer #3: No

**Data Deposition**

http://datadryad.org/submit?journalID=pgenetics&manu=PGENETICS-D-20-00971R2

**Press Queries**

---

## [Editor Report · Acceptance letter]

20 May 2021

PGENETICS-D-20-00971R2 

Variation in phenotypes from a Bmp-Gata3 genetic pathway is modulated by Shh signaling 

Dear Dr Swartz, 

We are pleased to inform you that your manuscript entitled "Variation in phenotypes from a Bmp-Gata3 genetic pathway is modulated by Shh signaling" has been formally accepted for publication in PLOS Genetics! Your manuscript is now with our production department and you will be notified of the publication date in due course.

With kind regards,

Katalin Szabo

PLOS Genetics

On behalf of:
